# Human Arboviral Infections in Italy: Past, Current, and Future Challenges

**DOI:** 10.3390/v15020368

**Published:** 2023-01-27

**Authors:** Benedetta Rossi, Filippo Barreca, Domenico Benvenuto, Neva Braccialarghe, Laura Campogiani, Alessandra Lodi, Camilla Aguglia, Rosario Alessandro Cavasio, Maria Laura Giacalone, Dimitra Kontogiannis, Martina Moccione, Vincenzo Malagnino, Massimo Andreoni, Loredana Sarmati, Marco Iannetta

**Affiliations:** 1Infectious Disease Clinic, Policlinico Tor Vergata University Hospital, Viale Oxford 81, 00133 Rome, Italy; 2Department of System Medicine Tor Vergata, University of Rome, Via Montpellier 1, 00133 Rome, Italy

**Keywords:** arbovirus, climate, Toscana, Usutu, tick-borne encephalitis, sandfly fever Naples virus, sandfly fever Sicilian virus, Chikungunya, West Nile, outbreak

## Abstract

Arboviruses represent a public health concern in many European countries, including Italy, mostly because they can infect humans, causing potentially severe emergent or re-emergent diseases, with epidemic outbreaks and the introduction of endemic circulation of new species previously confined to tropical and sub-tropical regions. In this review, we summarize the Italian epidemiology of arboviral infection over the past 10 years, describing both endemic and imported arboviral infections, vector distribution, and the influence of climate change on vector ecology. Strengthening surveillance systems at a national and international level is highly recommended to be prepared to face potential threats due to arbovirus diffusion.

## 1. Introduction

The acronym arbovirus derives from arthropod-borne viruses and refers to a variety of RNA viruses that share the unique characteristic of being transmitted to vertebrate hosts by hematophagous arthropods, including ticks, mosquitos, and sandflies [1].

Arboviruses represent a public health concern in Italy, mostly because they can infect humans, causing potentially severe emergent or re-emergent diseases. Furthermore, arboviruses have already been the cause of epidemic outbreaks in the last few decades in Italy, with the potential of introducing new species previously confined to tropical and sub-tropical regions [2]. Arboviral infections, such as those caused by West Nile Virus (WNV), Toscana Virus (TOSV), Sandfly Fever Sicily Virus (SFSV), Sandfly Fever Naples Virus (SFNV), Usutu Virus (USUV), and tick-borne encephalitis virus (TBEV), may be considered endemic in Italy; Chikungunya Virus (CHIKV), Dengue Virus (DENV), and Zika Virus (ZIKV) are under surveillance, as imported infections could lead to autochthonous outbreaks and subsequent endemic circulation in Italy [3]. Furthermore, global travel and climate changes may favor the circulation of arboviral infections [4].

Based on the evolving epidemiological situation and due to the increasing number of cases, the Italian Ministry of Health has activated a national surveillance system for arboviral diseases. This integrated surveillance system, coordinated by the Italian National Institute of Health (Istituto Superiore di Sanità, ISS) and the Zooprophylactic Institute of Abruzzo and Molise (IZS-AM), annually publishes bulletins to trace and tackle arboviral infections, to ensure early detection of autochthonous and imported cases and contain any possible spread [5].

In this narrative review, we aim to summarize the Italian epidemiology of arboviral infections over the past 10 years, to describe vector distribution in Italy, and possible new threats for the next future, also considering the influence of climate change.

## 2. Endemic Arboviruses in Italy

### 2.1. West Nile Virus and Usutu Virus

WNV and USUV are two flaviviruses whose diffusion in the Italian avian fauna has been well established for a long time. In recent decades, they have become relevant also for human health. WNV was isolated for the first time in Uganda in 1937 [6]. It encompasses five lineages, but only lineages 1 and 2 are associated with epidemics. Birds represent the main reservoir and become highly viremic when infected (amplifying host), allowing WNV transmission to mosquitoes of Culex species that feed on them. Infected mosquitoes can spread the virus to non-avian vertebrate hosts such as horses and humans, who develop a low-level viremia, usually insufficient to transmit the virus to other mosquitoes (dead-end host) [7] (Table 1).

In Italy, the first cases in horses were reported in 1998, while the first cases of the most severe form of WNV infection, West Nile neuroinvasive disease (WNND), were diagnosed in humans in 2008 [7,8,9]. Since 2008, an average of 60 human cases per year of WNV infections, including asymptomatic infection, febrile syndrome, and WNND, has been reported, with an increasing trend after 2011, presumably because of the introduction of lineage 2 in Italy [9]. Since 2018, a further increase in the number of cases has been reported, especially during summer. This phenomenon is sustained by both biotic factors, such as migratory birds that brought the virus to new regions and the high density of *Culex* mosquitoes, and abiotic factors, such as higher temperatures in spring and summer [10]. Sequencing of the WNV genome from infected humans, birds, and mosquitoes documented the persistence of the same viral strains in consecutive years in Italy, supporting the hypothesis of an overwintering capacity of WNV, rather than different introductions [11]. In 2022, there was a dramatic increase in the number of WNV cases, with 586 human infections reported by the European Centre for Disease Prevention and Control (eCDC) up to the 25th of November [12], and 588 human infections according to the ISS bulletin [13] up to the 2nd of November (Table 2). Most of the reported cases occurred in the Veneto region, where a new strain of WNV lineage 1 was introduced in 2021 and is currently co-circulating with WNV lineage 2. This strain of WNV lineage 1 seems to be associated with an increased rate of WNND [14].

Usutu virus (USUV) is an emerging flavivirus isolated in South Africa in 1959 and introduced in Tuscany, Italy, in 1996, retrieved from archived tissue samples from birds [15,16]. This arbovirus shares its transmission cycle with the phylogenetically related WNV, and recognizes *Culex* mosquitoes as the main vector, birds as the amplifier hosts, and mammals (including humans) as the dead-end hosts [15].

Although USUV has mostly been associated with diseases in birds, it can also infect mammals, including humans, with a wide spectrum of signs and symptoms, ranging from asymptomatic forms to severe neurological impairment, similar to the WNND [17] (Table 1). In 2009, the first two cases of confirmed human disease worldwide were reported in two immunosuppressed patients in Italy, both presenting with neuroinvasive disease [18,19]. In the following years, considering the implementation of surveillance systems, USUV was identified in an increasing number of susceptible hosts; it was detected in different native (mainly *Culex pipiens*) and invasive mosquito species (*Aedes albopictus*, *Ae. japonicus*), and a rise in USUV human infection notifications was observed in Europe [20]. In Italy, this led to the development of a WNV and USUV dedicated surveillance system in 2017. Between 2017 and 2018, USUV has been detected in five Italian regions (Emilia-Romagna, Friuli Venezia Giulia, Latium, Lombardy, and Veneto) [3]. Eight clinically relevant cases (only one with a neuroinvasive form) were identified in 2018 (Veneto) and two symptomatic cases in 2022 (Lombardy and Emilia-Romagna) [13,15] (Table 2). Although several lineages of USUV were identified (three African and five European) so far, four different lineages (Europe-1, 2, 3, and 4) seem to be prevalent in Italy. Considering USUV spatial spread in Europe, Italy seems to act mainly as a reservoir for USUV spread in neighboring countries, with two geographical clusters in northern and north-western regions [21].

USUV is often misidentified as WNV because of the cross-reactivity of serological assays, and new diagnostic tools are under development to improve diagnostic approaches [22,23]. Surveys conducted on blood donors and asymptomatic patients have shown a significant USUV seroprevalence in Italy, with some cases of detectable viremia in several Italian regions [13,22,24,25,26]; a higher USUV seroprevalence compared to WNV was found in the cerebrospinal fluid and sera of patients with neurologic impairment tested in Modena (Emilia-Romagna), suggesting an overall underreporting of the human USUV infection [27].

### 2.2. Toscana Virus, Sandfly Naples Virus, and Sandfly Sicilian Virus

Toscana Virus (TOSV) is a phlebovirus originally isolated in 1971 in Monte Argentario (Tuscany) from the sandflies Phlebotomus perniciosus and P. perfiliewi. [28].

It was recognized as the causative agent of neurological disease in humans only in 1983 [29]. Over the past two decades, circulation of TOSV has been detected mainly in the Mediterranean basin, typically between May and October, in humans and domestic animals [30]. A significant proportion of infections are asymptomatic; although, TOSV can reach the central nervous system (CNS) and cause acute aseptic meningitis, encephalitis, and meningoencephalitis. Other clinical manifestations associated with TOSV infection are described in Table 1. Severe forms with CNS involvement have been mainly reported in male adults, probably related to exposition during outdoor activities in the evening when phlebotomi are maximally active [30].

As described in Table 2, between June 2016 and October 2021, 331 confirmed cases of TOSV meningitis, meningoencephalitis, and encephalitis were reported in Italy, with 292 hospitalizations [31,32]. Even though earlier reports strongly confined TOSV infections within the Tuscany region, 61% of neuroinvasive cases occurred in Emilia Romagna, with an incidence rate two times higher than in other regions. This might be explained considering that Emilia-Romagna is located in the Po River Valley, which represents a fragile environment particularly vulnerable to climate variations. Indeed, changes in temperature and precipitation influence competent vector diffusion in this territory generating new favorable ecological niches for phlebotomi [33]. An elevated abundance of sandfly vectors has already been recorded in Emilia Romagna in previous studies on human leishmaniosis (phlebotomi are also vectors for Leishmania species) and further confirmed by recent entomological studies. The hilly environments in the central and eastern parts of the region are considered the most suitable areas to find abundant populations of sandflies [34,35].

Besides TOSV, two more sandfly-transmitted phleboviruses are endemic in Italy with a prevalent distribution in central and southern regions, namely, sandfly Naples virus (SFNV) and Sandfly Sicilian virus (SFSV) [36]. They were initially isolated in Italy, respectively, in Naples in 1944 and in Palermo (Sicily) in 1943; although, they are widely distributed in Southern Europe and other Mediterranean countries, reflecting the vectors’ distribution [37,38,39]. Phlebotomus perniciosus together with P. perfiliewi are the most efficient vectors for SFNV, whereas P. perniciosus and P. papatasi are able to transmit SFSV [40]. Classically, the symptomatic disease caused by SFNV and SFSV is a mild, self-resolving, flu-like illness, in which fever is always present (Table 1). No deaths associated with these phleboviruses have been reported, so far [41].

### 2.3. Tick-Borne Encephalitis Virus

Tick-born encephalitis (TBE) is an emerging zoonotic neurological disease caused by the tick-borne encephalitis virus (TBEV), which belongs to the flavivirus genus [42]. Among the known TBEV subtypes (European, Siberian, Far-Eastern, Baikalian, and Himalayan), the European one, which is the least virulent subtype, is the most diffuse in Italy and is transmitted by Ixodes ricinus [43]. The first cases of TBE in Italy were identified in Tuscany in the 1970s, whereas other sporadic foci were subsequently reported in the 1990s in the northern provinces of Trento (Trentino Alto Adige) and Belluno (Veneto), where the disease is now endemic [44]. As resumed in Table 1, clinical features range from asymptomatic or nonspecific flu-like symptoms to central nervous system complications in 20–30% of cases, such as severe meningoencephalomyelitis [45]. Italy is considered a low-incidence country for TBEV, mostly because of its restricted distribution to relatively small areas of the country, with sporadic cases in Emilia-Romagna, Tuscany, and Lazio. An endemic circulation is registered only in three main foci of the Triveneto area: the province of Trento (Trenino-Alto Adige), the pre-alpine Belluno province (Veneto), and the north-east of Friuli-Venezia Giulia. In this area, the number of cases has significantly increased over time, from 4.5 cases/year during the period 1975–2004 to 28 cases/year between 2000 and 2013 [46]. Although Italy is a country at low risk for TBE, the province of Belluno (Veneto), is classified by the World Health Organization (WHO) as a highly endemic area (>5 cases per 100,000 population), with an annual incidence of 5.95 cases per 100,000 population, according to the notifications received in the period 2015–2018 [44] (Table 2). A TBE vaccine is currently available for human use in Italy. Although official data on TBE vaccination rates in Italy are lacking, some reports suggest that vaccination coverage is low (10–40%), even in high-risk groups in highly endemic areas [44].

## 3. Imported Arboviruses and Outbreaks in Italy

Imported cases of infections due to arbovirus are commonly reported among travelers returning from endemic regions (Table 2). Moreover, in non-endemic areas where potentially competent vectors are present, the risk of autochthonous transmission after imported cases is relatively high. The continuous expansion of areas with arbovirus circulation, together with climate changes and the dispersion of competent vectors may increase the risk of outbreaks also in regions previously considered temperate climate areas [47].

### 3.1. Chikungunya Virus

CHIKV is a togavirus endemic in regions of Africa, South-East Asia, and in the Indian subcontinent, which has become established in the tropical Americas, and is transmitted by Aedes mosquitos. In Italy, the first cases were reported from July to September 2007 in Emilia Romagna, during an outbreak originating from a visitor from India, and subsequently sustained by autochthonous spread, with a total of 217 human cases and one death [48,49,50]. At that time, Aedes albopictus, which is a competent vector for CHIKV, was already established and abundant enough to sustain transmission following the introduction of the virus by an infected returning traveler. After this outbreak, only sporadic imported cases were reported in Italy, until 2013 (Table 2). In December 2013, an outbreak of CHIKV in the Western Hemisphere occurred in the Caribbean leading to the wide spread of the virus through the American continent. The virus was then imported into Europe by several viremic patients traveling from endemic countries [50]. During 2014, the number of imported cases of CHIKV registered in Italy increased up to 37 cases, including 18 cases connected to the Caribbean outbreak. In the following years, few imported cases were observed (Table 2). In 2017, a second autochthonous outbreak occurred in Italy, with 489 notified cases (270 confirmed and 219 probable), almost all confined to Latium and Calabria territories. It involved four outbreaks (Anzio, Latina, Rome, and Guardavalle Marina) sustained by a virus belonging to the Indian Ocean Lineage (IOL), with a hospitalization rate of 6% and one death [51,52]. This was the first known transmission of chikungunya in central and southern Italy [53]. In 2019, 18 cases of imported CHIKV from endemic regions (India 8%, Myanmar 23%, Congo 8%, Maldives 23%, and Brazil 38%) were registered [54]. In 2021, no cases of CHIKV were notified by the national surveillance of arboviral diseases, probably because of restriction measures adopted for the COVID-19 pandemic.

### 3.2. Dengue Virus

Dengue virus (DENV) is a flavivirus, responsible for the majority of febrile illnesses in tourists traveling in endemic areas, where it is mainly transmitted by Aedes aegypti [55] (Table 1). Travel-related cases of dengue fever in Europe reflect the evolution of dengue epidemiology in tropical regions, where the disease is endemic [56]. In Europe, DENV is not endemic despite the presence of competent vectors (Ae. albopictus), which may concur to establish new autochthonous cycles of infection [55]. In 2019, the highest number of imported cases of DENV (185) was observed in Italy. The subsequent decrease in the number of travel-related cases of dengue in Europe in 2020 (only 19 imported cases in Italy) can be explained by the implementation of restriction measures during the COVID-19 pandemic. The numbers of imported DENV cases per year in Italy (2010–2021) are summarized in Table 2. Moreover, all four DENV serotypes (1, 2, 3, and 4) have been identified in returning travelers in Italy [47]. No autochthonous dengue cases were described in Italy until 2020 [57]. During the summer of 2020, an autochthonous outbreak of DENV was documented in Italy, specifically in the Veneto region, with 11 autochthonous cases derived from a single imported case (index), of a young woman returning from Indonesia. Serotype 1 (DENV-1) was responsible for this outbreak [58].

### 3.3. Zika Virus

Zika virus (ZIKV) is a flavivirus transmitted to humans by Aedes mosquitos. Viremic travelers may cause autochthonous circulation after entering countries such as Italy, where competent vectors are present (for example, Ae. albopictus). Perinatal transmission and human-to-human transmission through blood transfusion and sexual intercourse have also been reported [59]. The first two cases of ZIKV infections imported in Italy were described in returning travelers from French Polynesia (January, 2014) [60]. ZIKV was previously limited to sporadic cases in Africa and Asia. In 2015, Zika virus caused an outbreak in Brazil, which involved 1,300,000 people, then spread rapidly throughout tropical South and Central America, and several imported cases were described in the USA and European countries, including Italy (102 cases in 2016) [59,61]. A few autochthonous cases of ZIKV have been documented in Italy, probably related to sexual rather than vector-related transmission [62,63] (Table 2).

## 4. Vectors and Climate Changes: The Italian Situation

Multiple anthropogenic factors, including land-use changes and global trade, together with warming and precipitation changes can influence the abundance of vectors and reservoir hosts for several arboviruses, thus exacerbating the risk of vector-borne (VBDs) and tick-borne diseases (TBDs). Indeed, in Italy in the last decades, several VBDs have emerged unexpectedly because climate changes have already contributed to generate suitable conditions for stable arthropod vector replication and arbovirus transmission [4].

### 4.1. Mosquitos

Mosquitos are small flies belonging to the family of Culicidae, which play a primary role in arbovirus transmission because of their widespread diffusion. In Italy, both invasive (Aedes) and native species (Culex) are present.

#### 4.1.1. *Aedes*

According to the current surveillance system, five invasive *Aedes* mosquito species have been identified as having steadily established in Europe, namely, *Ae. albopictus*, *Ae. aegypti*, *Ae. japonicus*, *Ae. Atropalpus*, and *Ae. koreicus*. Among this group of culicidae, the two most prevalent species representing a public health issue in Europe are *Ae. aegypti* and *Ae. albopictus* [64].

*Ae. aegypti* evolved in sub-Saharan Africa, and appeared for the first time in many southern European countries and the Americas at the end of the 1700s, imported through the slave trade. Although *Ae. aegypti* disappeared from temperate zones during winter because it does not enter into diapause and cannot survive the winter, subsequent reintroductions in temperate zones were due to sailing ships during summer. During the 1900s, an expansion outside the original geographical area was observed, mainly because of trades and troop movements during World War II. Nevertheless, *Ae. aegypti* is still not present in Italy [65]. *Ae. aegypti* has an increased ability to spread diseases compared to *Ae. albopictus*, considering its feeding patterns, characterized by several human blood meals from multiple individuals in a short period of time. Moreover, *Ae. aegypty* has better vector competence for chikungunya (CHIKV) and dengue (DENV) viruses compared to *Ae. Albopictus*, and it is also the principal vector for Yellow Fever Virus (YFV) [66].

*Ae. albopictus*, commonly called the Asian tiger mosquito, is native to tropical forests in South Asia. It is already established in most regions of southern Europe, with Albania and Italy being the first colonized countries in 1979 and 1990, respectively. *Ae. albopictus* spread was related to tires and ornamental plants containing mosquito eggs and exported from endemic areas of Asia. *Ae. albopictus* is a competent vector for CHIKV, DENV, and filarial parasites (*Dirofilaria* species, which primarily parasites dogs but can also affect humans). Experimental studies have demonstrated that *Ae. albopictus* can also be a competent vector for West Nile Virus [66]. Maps describing current *Ae. aegypti* and *Ae. albopictus* distributions in Europe are also available [67,68].

Both *Aedes* species are highly anthropophilic and daytime biters, with peak activity at dawn and dusk. They compete for the same habitats, sharing mostly urban and peri-urban areas; although, *Ae. albopictus* has a cooler thermal optimum (26 °C) than *Ae. aegypti* (29 °C), and is more ecologically flexible, living also in rural, residential, and agricultural habitats. For these reasons *Ae. albopictus* is frequently found in temperate zones [69,70]. Furthermore, it bites humans and a wide range of mammals and birds that are not susceptible to the arboviruses they carry. Conversely, *Ae. aegypti* exclusively feeds on humans.

A new mosquito species native to South Asia, *Aedes koreicus*, was detected in Italy in 2011, and in other European countries such as Belgium, Switzerland, Germany, Hungary Slovenia, and Russia [71]. Considering its ability to adapt to temperate climates, it can compete with *Ae. albopictus*; although, it seems unlikely that *Ae. koreicus* could replace *Ae. albopictus* [72]. In northern Italy, *Ae. koreicus* has also colonized mountainous districts in the Veneto region, where *Ae. albopictus* remains absent. With the establishment of this new mosquito, humans and other mammals could face new potential threats, considering that experimental data have demonstrated the ability of *Ae. koreicus* to transmit Japanese encephalitis virus (JEV), and two other already endemic flaviviruses in the Veneto region: WNV and USUV [69,71,72].

#### 4.1.2. *Culex*

The genus *Culex* is of high medical and veterinary interest due to its widespread distribution; it encompasses 768 species divided among 26 subgenera able to transmit a wide range of pathogens, causing diseases in both animals and humans [73].

Mosquitoes in the *Cx. pipiens* complex (or assemblage) are native to Africa, Asia, and Europe; although, they are currently distributed worldwide. *Cx. pipiens* complex includes different taxa: *Cx. pipiens* and *Cx. quinquefasciatus*, mostly distributed in temperate and tropical areas, as well as *Cx. australicus* and *Cx. globocoxitus*, only present in Australia. The taxon *Cx. pipiens* has two different biotypes: the *Cx. pipiens pipiens*, and the *Cx. pipiens molestus*. The two biotypes differ in behavioral and ecological characteristics: the *pipiens* biotype, also known as the rural form, is highly ornithophilic, even though several studies also demonstrated potential anthropophilic behavior [74]; it lays eggs in open-air spaces and the females hibernate during winter (diapause). The *molestus* biotype, also known as the urban form, is anthropophilic even though it can also bite other mammals; it does not require large habitat for oviposition and does not diapause [75].

In Italy, *Cx. pipiens* mosquitoes are present and outspread, and they represent competent vectors for both WNV and USUV transmission [76]. Outside Europe, Culex mosquitos are also competent for Rift Valley Fever, (RVFV), Japanese encephalitis virus (JEV), Sindbis virus (SINV), and Tahyna virus (TAHV) [77,78], which can cause severe human diseases.

### 4.2. Phlebotomine Sandflies

Phlebotomine sandflies are responsible for the transmission of a few types of parasites between humans and nonhuman animal reservoirs and they represent important vectors for arboviral infections [75,79]. They belong to the Diptera order, Psychodidae family, and are small insects with sand-like color (hence, the name “sand flies”) [80]. Only the females are hematophagous and have no specific host preference since they mostly bite any warm-blooded vertebrate (the *Sergentomyia* genus feeds mainly on reptiles). Mating occurs after the blood meal or in the presence of a host to sting, and the eggs are subsequently laid. They live in domestic, peri-domestic, and wild environments, being generally active during twilight and night hours, avoiding sunlight, and hiding in cool and humid environments such as stables, houses, chicken coops, and cracks of walls, rock, and soil. Phlebotomine are quite spread in the Mediterranean basin and the Middle East, and colonization by passive transport from distant regions has never been reported [80].

Out of the over 800 described species of Phlebotomine, 8 species are endemic in Italy, and they belong to 2 genera: *Phlebotomus* and *Sergentomyia*. Seven species belong to the *Phlebotomus* genus (*P. perniciosus*, *P. perfiliewi*, *P. neglectus*, *P. ariasi*, *P. papatasi*, *P. sergenti*, and *P. mascitti*), and one species belongs to the *Sergentomyia* genus (*S. minuta*). This latter species feeds on cold-blooded animals and does not cause diseases in humans, but it can be an important ecological indicator of the presence of Phlebotomine [81]. The most widespread and most abundant species in Italy is *P. perniciosus*, an established vector for visceral leishmaniasis, and also competent for Toscana virus (TOSV) and Sandfly Fever Naples virus (SFNV) [82]. TOSV was isolated also in *P. perfiliewi*, while SFSV was isolated in *P. papatasi* [38,83].

### 4.3. Ticks

The most widespread and relevant species of ticks in Italy are *Ixodes ricinus* (the wood tick), *Rhipicephalus sanguineus* (the dog tick), *Hyalomma marginatum*, and *Dermacentor reticulatus* [84].

#### 4.3.1. Ixodes

Ticks from the Ixodidate family are vectors for several tick-borne flaviviruses such as tick-borne encephalitis virus (TBEV), Powassan virus (POWV), and Omsk hemorrhagic fever virus, and bunyaviruses such as Crimean-Congo hemorrhagic fever virus (CCHFV), all agents of human diseases that could lead to fatal complications such as viral hemorrhagic fever (VHF) [85,86]. In particular, *I. ricinus* acts both as vector and reservoir for several bacterial, viral, and protozoan agents causing animal and human diseases in Europe [87]. In Italy, it has been detected in most northern and central regions, especially in woods and shrubby habitats where the relative humidity allows the tick to complete its 3-year developmental cycle. It is the most competent vector for TBEV in Europe. As previously described, in recent years, there has been an increase in TBE cases in Italy. Rizzoli et al. have shown that this increase is not directly associated with changes in climate variables. Conversely, changes in the vegetation that make the habitat more suitable for small mammals, which represent TBEV reservoirs, seem to best predict the increase in TBE incidence [88]. A phylogenetic analysis of TBEV Italian isolates suggested that the different TBEV strains originated independently after different introductions from neighboring countries, presumably through migratory birds [43].

#### 4.3.2. Hyalomma

Ticks belonging to the *Hyalomma* genus, in which immature stages are strictly associated with birds, can potentially acquire WNV and USUV from an infected host and possibly act as vectors, as demonstrated experimentally [89]. Migratory birds can carry infected ticks, as well as non-endemic pathogens such as *Rickettsia africae*, from sub-Saharan Africa, into Europe [90]. With climate changes, the likelihood of the establishment of permanent *Hyalomma* populations in central and northern Europe is increasing [91].

### 4.4. The Impact of Climate Changes on VBDs and TBDs

VBDs and TBDs are highly climate-sensitive because the vectors are poikilotherms (body temperature fluctuates accordingly to environmental temperature), and the rate of most physiological activities such as digestion of blood meals, egg-laying, and refeeding is regulated by ambient temperature [92]. Additionally, vector ranges, distribution, and seasonality are already altered because of changes in weather conditions, particularly temperatures, precipitation, and humidity. In its sixth assessment report, the Intergovernmental Panel on Climate Change (IPCC) stated with high confidence that the prevalence of vector-borne diseases has increased in recent decades. Arboviral infections such as dengue and West Nile, together with malaria and Lyme disease, are expected to further increase over the next 80 years if measures are not taken to adapt and strengthen control strategies [93]. Understanding the extent of the influence of climate change on the distribution and frequency of arthropod vectors and arboviral infection is still challenging, considering the high number of factors with complex intertwine that can contribute to these changes. Moreover, it is difficult to distinguish natural climate variability from anthropogenic-driven environmental change. Despite these complexities, the different aspects of vector-borne disease, including pathogens, vectors, and reservoir hosts, are highly responsive to environmental modifications. Indeed, changes in the rates of vector-borne diseases at given locations are often associated with concomitant changes in the local environmental conditions [4].

## 5. Conclusions and Future Directions

Arthropods and viruses are evolving organisms that react to ecological and anthropogenic factors, in an attempt to adapt to a modified environment with new selective pressures. Global concern about the spread of arboviruses to new geographical areas is rising. In Italy and in other regions of southern Europe, the possibility of autochthonous transmission of non-endemic arboviral infections, such as dengue and chikungunya, has already been documented. Moreover, the global spread of arthropods together with a continuous intensification of human movements can contribute to the introduction of new vectors and tick-borne diseases in previously unaffected countries. One example could be the possible introduction of the Crimean-Congo hemorrhagic fever virus (CCHFV) in Italy, through migratory birds and ticks. Indeed, CCHFV human infections have already been identified in the former Soviet Union and some Balkan countries and more recently in western Spain (first two cases in 2016). Since 2016, at least 10 confirmed cases of human infection have been reported in Spain, and studies on ticks have demonstrated that CCHFV is widespread in the country [94]. Considering that CCHFV has extended from Eastern to Western Europe, it is reasonable to hypothesize that this virus could reach Italy.

Countries that potentially have the conditions for the introduction of arthropod-related emerging diseases need to strengthen surveillance systems to face these new threats.

## Figures and Tables

**Table 1 viruses-15-00368-t001:** Epidemiological and clinical characteristics of arboviral infections in Italy.

Arbovirus	Vector	Type of Circulation	Clinical Features	Severe Disease
West Nile virus (WNV)	***Culex pipiens***, *Aedes koreicus*	endemic, imported	West Nile Fever: fever, headache, nausea and vomiting, lymphadenopathies, rash	West Nile neuroinvasive disease: meningoencephalitis with disorientation, tremors, impaired vision, numbness, convulsions, paralysis, and coma
Usutu virus (USUV)	***Culex pipiens***, *Aedes albopictus*	endemic	fever, headache, arthromyalgias, nausea and vomiting, lymphadenopathies, rash	high fever, stiff neck, confusion, disorientation, coma, seizures, loss of vision, numbness, and paralysis.
Toscana virus (TOSV)	*Phlebotomus perniciosus*, *Phlebotomus perfiliewi*	endemic	flu-like symptoms	meningoencephalitis, sensory polyradiculopathy, Guillain–Barrè syndrome, testicular involvement, myositis, and fasciitis
Sandfly Fever Sicilian virus (SFSV)	*Phlebotomus perniciosus;* *Phlebotomus papatasi*	endemic	flu-like symptoms	sporadic neuroinvasive disease reported
Sandfly Fever Naples virus (SFNV)	*Phlebotomus perniciosus*, *Phlebotomus perfiliewi*	endemic	flu-like symptoms	uncommon
Tick-borne encephalitis virus (TBEV)	***Ixodes ricinus***, *Ixodes persulcatus*, *Dermacentor*, *Haemaphysalis*	endemic, imported	high fever, severe headache, sore throat, tiredness, arthromyalgias	encephalitis, flaccid paralysis
Chikungunya virus (CHIKV)	***Aedes aegypti***, *Aedes albopictus*	imported, outbreaks	fever, joint pain	rarely ophthalmological, neurological and heart complications
Dengue virus (DENV)	***Aedes aegypti***, *Aedes albopictus*	imported, outbreaks	severe headache, eye pain, arthromyalgias, nausea and vomiting, rash	shock, internal bleeding, and even death. More common in people with previous DENV infection.
Zika virus (ZIKV)	***Aedes aegypti***, *Aedes albopictus*, *Culex pipiens*	imported	maculopapular rash, arthromyalgias, headache, and conjunctivitis.	Rarely Guillain–Barrè syndrome and neuroinvasive disease. Infection during pregnancy: teratogenic effects.

Vectors in bold represent the most competent vector.

**Table 2 viruses-15-00368-t002:** Incidence of endemic and imported arbovirosis in Italy 2010–2022.

	2010	2011	2012	2013	2014	2015	2016	2017	2018	2019	2020	2021	2022 *
*Endemic arbovirosis*													
**WNV**	3 ^a^	14 ^b^	50 ^b^	69 ^b^	24 ^b^	60 ^b^	76 ^b^	57 ^b^	576 ^b^	53 ^b^	66 ^b^	55 ^b^	588 ^b^
**USUV**	NA	NA	NA	NA	NA	NA	NA	NA	8 ^c^	1 ^a^	1 ^a^	2 ^a^	2 ^d^
**TOSV** *neuroinvasive*	NA	NA	NA	NA	NA	NA	94 ^a^	NA	89 ^a^	56 ^a^	36 ^a^	56 ^a^	100 ^a^
**TBEV**	NA	NA	NA	NA	NA	5 ^b^	48 ^b^	24 ^b^	39 ^b^	37 ^b^	55 ^b^	14 ^a^	40 ^a^
*Imported arbovirosis*													
**DENV**	50 ^a^	47 ^a^	79 ^a^	142 ^a^	79 ^a^	105 ^a^	106 ^a^	94 ^a^	108 ^a^	185	19 ^a^ (30) ^a^	11 ^a^	114 ^a^
**ZKV**	0 ^a^	0 ^a^	0 ^a^	0 ^a^	3 ^a^	4 ^a^	102 ^a^	26 ^a^	1 ^a^	4 ^a^	3 ^a^	0 ^a^	1 ^a^
**CHIKV**	7 ^a^	2 ^a^	5 ^a^	3 ^a^	37 ^a^	8 ^a^	17 ^a^	5 ^a^ (489) ^e^	5 ^a^	18 ^a^	3 ^a^	0 ^a^	0 ^a^

West Nile Virus (WNV), Usutu Virus (USUV), Toscana Virus (TOSV), Tick-Borne encephalitis Virus (TBEV), Dengue virus (DENV), Zika virus (ZKV), Chikungunya virus (CHIKV). The numbers in brackets represent the outbreaks. For TOSV, only neuroinvasive infections are reported. * 2022 data include cases up to October 3rd; ^a^ Epicentro ISS Data; ^b^ ECDC Surveillance and Disease Data; ^c^ Pacenti M et al. [15]; ^d^ ISS, IZS Teramo. Integrated Surveillance of West Nile and Usutu virus. Bulletin N. 20 of 2 November 2022; ^e^ ISS. Italy: Autochthonous Outbreaks of Chikungunya Virus Infection.

## Data Availability

Not applicable.

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
