# Peer review of "Human Arboviral Infections in Italy: Past, Current, and Future Challenges"

_viruses, 2023, doi:10.3390/v15020368_

Round 1

Reviewer 1 Report

Table 1. Zika virus can cause a variety of fetal effects in addition to microcephaly. Because of space limitations, simply indicating "teratogenic effects" would cover the range of effects included in the Congenital Zika Syndrome.  

Lines 127 -131. Is there published evidence that Phlebotomine vectors of TOSV are more abundant in the Po Valley than elsewhere?

Line 163. It would be worth noting that there are commercially available TBE vaccines available for human use. 

Lines 180-181. ... and in the Indian subcontinent, has become established in the tropical Americas, and is transmitted by ,,,

Lines 181-183 It would be good to mention that populations of the competent CHIK virus vector, Aedes albopictus, were already established and abundant enough to sustain transmission following  introduction of the virus by the infected Indian traveler. 

3.2 Dengue viruses. It would be of interest to mention which of the four dengue virus serotypes have been introduced into Italy if that information is available.

Line 223 Zika virus in Brazil then spread rapidly throughout tropical South and Central America.

Line 267. The statement that Aedes aegypti disappeared for unknown reasons is vague and inaccurate for many localities. Disappeared from where? It disappeared from temperate zones because they do not enter into diapause, hence do not survive the winter and were reintroduced during following summers by sailing ships. They have not disappeared in the tropics around the world. 

Line 258. eggs were introduced in tires and some ornamental plants. Also, it would be worth mentioning that the ECDC has a good map showing the current distribution of Ae. albopictus (see https://www.ecdc.europa.eu/en/publications-data/aedes-albopictus-current-known-distribution-march-2021) and Ae. aegypti (see https://www.ecdc.europa.eu/en/publications-data/aedes-aegypti-current-known-distribution-march-2021)

Line 263. Aedes aegypti exclusively feeds on humans. In addition to humans, Aedes albopictus also feeds on a variety of mammals that are not susceptible to the arboviruses they carry. This is an epidemiologically important difference.

Line 380. It would be worth noting that ticks infected with CCHF virus were first found in in western Spain in 2010. Since 2013 there have been 12 confirmed human cases of CCHF there, indicating that this virus has been established in Europe.  It is reasonable to suggest that this virus could reach Italy. 

Author Response

Comments and Suggestions for Authors

Table 1. Zika virus can cause a variety of fetal effects in addition to microcephaly. Because of space limitations, simply indicating "teratogenic effects" would cover the range of effects included in the Congenital Zika Syndrome. 

We agree with the reviewer and accordingly modified the text in table 1.

Lines 127 -131. Is there published evidence that Phlebotomine vectors of TOSV are more abundant in the Po Valley than elsewhere?

We thank the reviewer for this comment. Several published papers demonstrated a higher abundance of Phlebotomine vectors of TOSV in the Po Valley (which includes Piedmont, Lombardy, Emilia-Romagna and Veneto regions). Some of these studies aimed to investigate human leishmaniasis epidemiology in Northern Italian regions. We referred also to this type of literature, considering that TOSV and Leishmania share the same principal vectors, represented by sandflies. We added a sentence in the text to clarify this point (lines 131-136):

“An elevated abundance of sandfly vectors had already been recorded in Emilia Romagna in previous studies on human leishmaniosis (phlebotomi are also vectors for Leishmania species) and further confirmed by recent entomological studies. The hilly environments in the central and eastern part of the region are considered the most suitable areas to find abundant populations of sandflies [34,35].”

Line 163. It would be worth noting that there are commercially available TBE vaccines available for human use.

We thank the reviewer for the comment. We added the following sentence in the text (lines: 168-170):

“A TBE vaccine is currently available for human use in Italy. Although official data on TBE vaccination rates in Italy are lacking, some reports suggest that vaccination coverage is low (10-40%), even in high-risk groups in highly endemic areas [44].” 

Lines 180-181. ... and in the Indian subcontinent, has become established in the tropical Americas, and is transmitted by ,,,

We modified the text as suggested by the reviewer (line 189).

Lines 181-183 It would be good to mention that populations of the competent CHIK virus vector, Aedes albopictus, were already established and abundant enough to sustain transmission following introduction of the virus by the infected Indian traveler.

 As suggested by the reviewer, we added the following sentence (lines: 193-195):

“At that time, Aedes albopictus, which is a competent vector for CHIKV, was already established and abundant enough to sustain transmission following introduction of the virus by an infected returning traveler.”

3.2 Dengue viruses. It would be of interest to mention which of the four dengue virus serotypes have been introduced into Italy if that information is available.

We thank the reviewer for this observation. All four DENV serotypes have been identified in returning travelers in Italy. Conversely, the outbreak in 2020 was sustained by DENV-1 serotype, which was identified in the index case and in all the derived autochthonous cases. We added two sentences in the text to clarify these points (lines 222-223 and 226-227).

“Moreover, all four DENV serotypes (1, 2, 3 and 4) have been identified in returning travelers in Italy [47].”

“Serotype 1 (DENV-1) was responsible of this outbreak [58].”

Line 223 Zika virus in Brazil then spread rapidly throughout tropical South and Central America.

The text has been modified according to reviewer’s suggestions (235-238):

“In 2015, Zika virus caused an outbreak in Brazil, which involved 1,300,000 people, then spread rapidly throughout tropical South and Central America and several imported cases have been described in USA and European countries, including Italy (102 cases in 2016) [59,61].”

Line 247. The statement that Aedes aegypti disappeared for unknown reasons is vague and inaccurate for many localities. Disappeared from where? It disappeared from temperate zones because they do not enter into diapause, hence do not survive the winter and were reintroduced during following summers by sailing ships. They have not disappeared in the tropics around the world.

We thank the reviewer for the comment. We modified the paragraph as follows (261-263):

“Although Ae. aegypti disappeared from temperate zones during winter because it does not enter into diapause and cannot survive the winter, subsequent reintroductions in temperate zones were due to sailing ships during summer.”

Line 258. eggs were introduced in tires and some ornamental plants. Also, it would be worth mentioning that the ECDC has a good map showing the current distribution of Ae. albopictus (see https://www.ecdc.europa.eu/en/publications-data/aedes-albopictus-current-known-distribution-march-2021) and Ae. aegypti (see https://www.ecdc.europa.eu/en/publications-data/aedes-aegypti-current-known-distribution-march-2021)

We thank the reviewer for the comment. We modified the text according to the suggestions (lines 274-276 and 279-280):

“Ae. albopictus spread was related to tires and ornamental plants containing mosquitos’ eggs and exported from endemic areas of Asia.”

“Maps describing current Ae. aegypti and Ae. albopictus distribution in Europe are also available [67,68]”

Line 263. Aedes aegypti exclusively feeds on humans. In addition to humans, Aedes albopictus also feeds on a variety of mammals that are not susceptible to the arboviruses they carry. This is an epidemiologically important difference.

The text has been modified according to reviewer’s suggestions (lines 286-288):

“Furthermore, it bites humans and a wide range of mammals and birds that are not susceptible to the arboviruses they carry. Conversely, Ae. aegypti exclusively feeds on humans.”

Line 380. It would be worth noting that ticks infected with CCHF virus were first found in in western Spain in 2010. Since 2013 there have been 12 confirmed human cases of CCHF there, indicating that this virus has been established in Europe.  It is reasonable to suggest that this virus could reach Italy.

We thank the reviewer for the comment. We added the following sentence to the text (lines 400-405):

“Indeed, CCHFV human infections have already been identified in former Soviet Union and some Balkan countries and more recently in western Spain (first two cases in 2016). Since 2016 at least 10 confirmed cases of human infections have been reported in Spain, and studies on ticks have demonstrated that CCHFV is widespread in the Country [95]. Considering that CCHFV has extended from Eastern to Western Europe, it is reasonable to hypothesize that this virus could reach Italy.”

Reviewer 2 Report

This is a nicely written review, well organized, and well researched. The authors were clearly focused on Italy, and perhaps neglected some important references because they were specifically from Italy, but this is not so important. The authors made some mistakes, both in factual statements and in English writing/grammar, that should be corrected. And actually the factual errors may be due to poor/ambiguous writing.

Author Response

Comments and Suggestions for Authors

This is a nicely written review, well organized, and well researched. The authors were clearly focused on Italy, and perhaps neglected some important references because they were specifically from Italy, but this is not so important. The authors made some mistakes, both in factual statements and in English writing/grammar, that should be corrected. And actually the factual errors may be due to poor/ambiguous writing.

We thank the reviewer for the comments. We have revised the text and amended the mistakes.